# Transmission Patterns of Co-Circulation of Omicron Sub-Lineages in Hong Kong SAR, China, a City with Rigorous Social Distancing Measures, in 2022

**DOI:** 10.3390/v16060981

**Published:** 2024-06-18

**Authors:** Ning Chow, Teng Long, Lam-Kwong Lee, Ivan Tak-Fai Wong, Annie Wing-Tung Lee, Wing-Yin Tam, Harmen Fung-Tin Wong, Jake Siu-Lun Leung, Franklin Wang-Ngai Chow, Kristine Shik Luk, Alex Yat-Man Ho, Jimmy Yiu-Wing Lam, Miranda Chong-Yee Yau, Tak-Lun Que, Kam-Tong Yip, Viola Chi-Ying Chow, River Chun-Wai Wong, Bobo Wing-Yee Mok, Hong-lin Chen, Gilman Kit-Hang Siu

**Affiliations:** 1Department of Health Technology and Informatics, The Hong Kong Polytechnic University, Kowloon, Hong Kong Special Administrative Region, China; ning-nico.chow@polyu.edu.hk (N.C.); lamkwongeddie.lee@connect.polyu.hk (L.-K.L.); tak-fai.wong@connect.polyu.hk (I.T.-F.W.); wing-tung-annie.lee@polyu.edu.hk (A.W.-T.L.); wing-yin10.tam@polyu.edu.hk (W.-Y.T.); fung-tin.wong@polyu.edu.hk (H.F.-T.W.); sl-jake.leung@connect.polyu.hk (J.S.-L.L.); franklin.chow@polyu.edu.hk (F.W.-N.C.); 2Centre for Virology, Vaccinology and Therapeutics Limited, The University of Hong Kong, Hong Kong Special Administrative Region, China; longt@hku.hk (T.L.); bobomok@hku.hk (B.W.-Y.M.); hlchen@hku.hk (H.-l.C.); 3Department of Microbiology and State Key Laboratory for Emerging Infectious Diseases, School of Clinical Medicine, Li Ka Shing Faculty of Medicine, The University of Hong Kong, Hong Kong Special Administrative Region, China; 4Department of Pathology, Princess Margaret Hospital, Hong Kong Special Administrative Region, China; sluk@ha.org.hk (K.S.L.); hym473@ha.org.hk (A.Y.-M.H.); 5Department of Clinical Pathology, Pamela Youde Nethersole Eastern Hospital, Hong Kong Special Administrative Region, China; lyw543a@ha.org.hk (J.Y.-W.L.); ycy801@ha.org.hk (M.C.-Y.Y.); 6Department of Clinical Pathology, Tuen Mun Hospital, Hong Kong Special Administrative Region, China; quetl@ha.org.hk (T.-L.Q.); yipkt@ha.org.hk (K.-T.Y.); 7Department of Microbiology, Prince of Wales Hospital, Hong Kong Special Administrative Region, China; chowcyv@ha.org.hk (V.C.-Y.C.); wcw372@ha.org.hk (R.C.-W.W.)

**Keywords:** COVID-19, Omicron, co-circulation, social distancing, quarantine arrangements

## Abstract

Objective: This study aimed to characterize the changing landscape of circulating SARS-CoV-2 lineages in the local community of Hong Kong throughout 2022. We examined how adjustments to quarantine arrangements influenced the transmission pattern of Omicron variants in a city with relatively rigorous social distancing measures at that time. Methods: In 2022, a total of 4684 local SARS-CoV-2 genomes were sequenced using the Oxford Nanopore GridION sequencer. SARS-CoV-2 consensus genomes were generated by MAFFT, and the maximum likelihood phylogeny of these genomes was determined using IQ-TREE. The dynamic changes in lineages were depicted in a time tree created by Nextstrain. Statistical analysis was conducted to assess the correlation between changes in the number of lineages and adjustments to quarantine arrangements. Results: By the end of 2022, a total of 83 SARS-CoV-2 lineages were identified in the community. The increase in the number of new lineages was significantly associated with the relaxation of quarantine arrangements (One-way ANOVA, F(5, 47) = 18.233, *p* < 0.001)). Over time, Omicron BA.5 sub-lineages replaced BA.2.2 and became the predominant Omicron variants in Hong Kong. The influx of new lineages reshaped the dynamics of Omicron variants in the community without fluctuating the death rate and hospitalization rate (One-way ANOVA, F(5, 47) = 2.037, *p* = 0.091). Conclusion: This study revealed that even with an extended mandatory quarantine period for incoming travelers, it may not be feasible to completely prevent the introduction and subsequent community spread of highly contagious Omicron variants. Ongoing molecular surveillance of COVID-19 remains essential to monitor the emergence of new recombinant variants.

## 1. Background

In 2022, many countries adopted the aim of living with the virus, whereas the Hong Kong government maintained stringent contact tracing and social distancing practices to keep COVID-19 cases at low levels. This strategy conferred the community seven months of nearly zero local cases in 2021 [1]. However, in January 2022, the fifth wave of local outbreak was kickstarted by Omicron variant BA.2.2, with more than one million self-testing or laboratory-confirmed infections from February to March 2022 recorded [2]. During the fifth wave, the government adjusted the social distancing policy and quarantine arrangements for inbound travelers several times. The number of confirmed local cases fluctuated and new lineages were found in the community. Several studies already elucidated the inverse relationship between the case number and the mobility of citizens in Hong Kong [3,4], while studies about the emergence of a new lineage from external sources such as visitors were limited. Therefore, in this study, we aimed to elucidate the relationship between the social distancing policy, quarantine arrangement and the dynamics of Omicron sub-lineages in the community in 2022.

## 2. Methods

### 2.1. Samples

A total of 4684 COVID-19-positive cases were collected from three public hospitals, namely Pamela Youde Nethersole Eastern Hospital (PYNEH), Tuen Mun Hospital (TMH) and Prince of Wales Hospital (PWH), during the whole of 2022. All these cases were defined as local cases as these patients did not have any travel history during the incubation period. All samples were de-identified before being sent to the laboratory for genomic surveillance. This study was approved by the Institutional Review Boards of The Hong Kong Polytechnic University (RSA20021) and these three hospitals (HKECREC-20200014; KWC-20200040; NTWC-20200038).

### 2.2. Whole-Genome Sequencing of SARS-CoV-2

The extraction of total nucleic acid from these respiratory specimens was performed using NucliSENS^®^ easyMAG^®^ (bioMérieux, Boxtel, The Netherlands) following the standard protocol from the manufacturers. The SAR-CoV-2 RNA were reverse-transcribed into cDNA using LunaScript™ RT SuperMix Kit (New England Biolabs, Hitchin, UK) following the manufacturer’s instructions. The viral cDNA was then amplified using the SARS-CoV-2-Midnight-1200 Amplicon Panel, and Q5^®^ High-Fidelity DNA Polymerase (New England Biolabs). The amplicons were quantified using the Qubit 2 fluorometer (Thermo Fisher Scientific, Waltham, MA, USA) prior to library preparation. The library preparation was performed following the protocol ‘PCR tiling of SARS-CoV-2 virus—rapid barcoding (SQK-RBK110.96)’ (version: PCTR_9125_v110_revH_24Mar2021-minion). The library was loaded and whole-genome sequenced on MinION or GridION (Oxford Nanopore Technologies, Oxford, UK) with R9.4.1 flow cells.

### 2.3. Analysis of Phylogeny and the Dynamics of Co-Circulation of Omicron Sub-Variants

The SARS-CoV-2 consensus genomes were constructed using wf-artic v0.3.21 [5]. Only genomes with coverage > 90% were included in the analysis. Multiple sequence alignment (MSA) was performed by Multiple Alignment using Fast Fourier Transform (MAFFT) with Wuhan-Hu-1 genome (NCBI accession number: MN908947.3) as the reference genome, and the options --6merpair, --keeplength, and --addfragments were used [6]. Lineage was identified using pangolin v4.2. The maximum likelihood phylogeny of the cases was determined using IQ-TREE with GTR+F+I+R3 as the best-fitting substitution model chosen by Bayesian Information Criterion (BIC) [7], while the dynamics of co-circulation of Omicron sub-variants in the community over the study period were illustrated by a time tree constructed using Nextstrain [8]. The number of confirmed cases and death cases were retrieved from DATA.GOV.HK [9]. The implementation dates and details of the quarantine arrangement in phases for the inbound travelers and social distancing policies were reviewed to investigate the change in the lineage distribution after each amendment.

### 2.4. Statistical Analysis

Correlation analysis of the number of lineages, death rate and the total days of quarantine was performed using the Spearman correlation coefficient or Pearson correlation coefficient accordingly. A one-way analysis of variance (one-way ANOVA) test was used to compare the number of new lineages and death rate between different quarantine arrangement models. A *p*-value of <0.05 was considered statistically significant.

## 3. Results

### 3.1. Overview of the Epidemic in 2022

Our genomic surveillance using 4684 local COVID-19 genomes revealed the emergence of 83 lineages in Hong Kong throughout 2022. Phylogenetic analysis of all sequences demonstrated that BA.2.2, which caused the flare-up of the epidemic, occupied the heaviest proportion. The remaining cases were categorized into the Delta variant AY.127 and four Omicron-related clusters (BA.1 sub-lineages, BA.2.12.1, BA.4 sub-lineages and BA.5 sub-lineages) (Figure 1). SARS-CoV-2 genomes might vary in length due to insertions or deletions (indels), which might influence the multiple sequence alignment result and subsequent phylogenetic analysis. Thus, we adopted Nextclade, another commonly used tool for multiple sequence alignment, to validate the results (Appendix A). The topology of two phylogenetic trees showed no difference, suggesting that length variation did not lead to significant inaccuracies in the phylogeny. Nextstrain was used to illustrate the timing of the occurrence of the lineages (Figure 2). At the beginning of the fifth wave, sporadic variants were detected in the community, but BA.2.2-related cases still constituted the majority until the middle of May, when the detected lineages became more diverse. In June, another wave of the epidemic was accompanied by the expansion of the BA.5 sub-lineage population. Ultimately, BA.5 sub-lineages outshone BA.2.2 and other co-circulating lineages to be the predominant strains. We quantified the weekly number of lineages throughout the whole of 2022, and a strong negative correlation was found between the weekly number of lineages and the duration of quarantine (Pearson *r* = −0.994, *p* = 0.0005, Appendix A and Appendix A). Association analysis of various quarantine arrangements models (Appendix A) and the dynamics of Omicron variants in the community revealed a significant association between the relaxation of quarantine arrangements for inbound travelers and an increase in the number of new lineages (One-way ANOVA, (F(5, 47) = 18.233, *p* < 0.001)) (Appendix A).

### 3.2. Change in the Dynamics of Circulating Lineages with the Adjustment of Quarantine Arrangement

The Hong Kong government implemented an up to 21-day mandatory hotel quarantine measurement for inbound travelers from the end of 2020. The monthly number of local confirmed cases from May to December 2021 was controlled at a near-zero level. In January 2022, a Pakistani woman who was infected by another inbound traveler during mandatory hotel quarantine sparked the Yat Kwai House-related cluster and triggered the local spread of Omicron variant BA.2.2 [10]. In that month, a total of 1554 tested-positive cases were reported, which was more than two-thirds of the total number of cases recorded in 2021 [11]. The government tightened social distancing measures and imposed ambush lockdowns, but the spread of the virus in the community was unmanageable. While the confirmed case number was surging, the government cut the mandatory hotel quarantine period by seven days (14 days of mandatory hotel quarantine and 7 days of medical surveillance at home, also known as the ‘14+7’ model) on 5 February, considering the short incubation period of Omicron variants. Finally, the number of confirmed cases peaked in early March 2022, with Omicron variant BA.2.2 as the only circulating strain (Figure 3 and Figure 4). On 1 April, the government adjusted the quarantine arrangement for inbound persons, reducing the duration of mandatory hotel quarantine from 14 days to 7 days, followed by 7-day self-monitoring (‘7+7’ model). The case number declined and plateaued at a low level in April and May (Figure 3). However, about seven weeks after the arrangement, new lineages started to emerge; four BA.2 sub-lineages, two BA.4 sub-lineages and sixteen BA.5 sub-lineages were found in the community (Figure 2).

The number of local cases steadily increased from May to August. The quarantine arrangement was then amended from ‘7+7’ to ‘3+4’ (3-day mandatory hotel quarantine and 4-day medical surveillance at home) on 12 August. Shortly after this amendment, the local case number suddenly exploded and reached the second peak (the sixth wave) in early September (Figure 3). One week later, BA.2.75.1 emerged in the community, followed by two other BA.2 sub-lineages, six BA.5 sub-lineages and XBD (Figure 2). The proportion of BA.5 continued to expand and finally outcompeted BA.2 sub-variants to become the dominant circulating Omicron variants in the community (Figure 4).

The number of local cases dropped from the peak and stayed at a relatively stable level at the end of September. The government then announced the cancellation of mandatory hotel quarantine and shortened the period of medical surveillance at home from four days to three days (‘0+3’ model). Prior to the cessation of this arrangement on 14th December, 35 new lineages, including 14 BA.2 sub-variants, 17 BA.5 sub-variants, 2 XBB sub-variants, XBC.1 and XBJ, were first detected in local cases (Figure 2). BA.2 sub-lineages were co-circulating with BA.5 sub-lineages, but the proportion of BA.2 sub-lineages continued to shrink, and that of BA.5 kept expanding (Figure 4). Of note, the level of case numbers during this period did not fluctuate significantly. No remarkable change in the death rate was observed when the number of cases climbed up (Spearman’s *ρ* = 0.258, *p* = 0.062, Appendix A and Appendix A). In addition, no significant association was observed between the relaxation of quarantine policy and death rate (one-way ANOVA, F(5, 47) = 2.037, *p* = 0.091) (Figure 4).

## 4. Discussion

To control the import of COVID-19 cases from other countries and prevent the spread of viruses in the local community, quarantine arrangements in phases for the inbound travelers and social distancing measurements were imposed from the emergence of the first case in Hong Kong. Since the latter half of 2021, Hong Kong experienced a seven-month period with nearly zero local cases due to the stringent quarantine policy (up to 21-day mandatory hotel quarantine) for inbound travelers implemented at the end of 2020. Unfortunately, such stringent measurements failed to prevent the fifth wave of the pandemic, which was triggered by a Pakistani woman, who got infected by another inbound traveler during mandatory hotel quarantine and brought the Omicron sub-variant BA.2 into the community in January 2022 [10]. Since then, the quarantine measurements have been adjusted several times regarding the pandemic situation. A total of four different quarantine arrangements for inbound travelers and social distancing measures were implemented sequentially during the whole of 2022 (Appendix A). Fluctuations in the local case number and emergence of new lineages were observed after each adjustment. The first adjustment was the replacement of the 21-day mandatory hotel quarantine with the ‘14+7’ model, which was implemented on 5 February. This amendment was followed by the peak of confirmed cases number in early March. At the end of February, the government implemented a vaccine pass arrangement to encourage the community to get vaccinated. Boosted vaccination coverage, together with previous natural infections, might have been the major cause of the subsided number of confirmed cases (Figure 3).

During the effective period of the ‘7+7’ model (1 April to 11 August 2022), the number of cases was successfully suppressed at a low level. However, the emergence of new lineages indicated that the seven-day-shortened mandatory quarantine period led to the penetration of new lineages brought by the imported cases into the community (Figure 2). Simultaneously, the government relaxed the social distancing measures, including the resumption of face-to-face classes and dinnertime dine-in service, as well as the re-opening of most previously closed premises (Appendix A). These factors may have contributed to the rebound in local cases in late May (Figure 3).

The sixth wave appeared in early September, when the ‘3+4’ model was implemented to replace the ‘7+7’ model. A significant increase in case number was accompanied by the expansion of the BA.5-infected population. This could be explained by several reasons. First, the shortened mandatory quarantine period may have allowed travelers to bring BA.5 sub-lineages into the community. The compulsory hotel quarantine period was shortened from seven days to three days. After three-day mandatory hotel quarantine, they were given an ‘amber code’ under the vaccine pass during the period of medical surveillance at home. Although the inbound travelers were not allowed to enter the premises subject to ‘active checking’ of the vaccine pass such as catering premises and fitness premises, they could take public transport and go to work or university. This provided a window to introduce the imported cases and new lineages into the community. Second, BA.5 was shown to have higher transmissibility and was better able to evade immunity from vaccines and previous infections with other variants [12]. The relaxed control measures, together with the natural advantages of BA.5, allowed the BA.5-infected population to expand rapidly compared to other Omicron variants.

From the end of September, when the ‘0+3’ model was launched, to the cessation of all quarantine measurements on 14 December, a pool of new lineages was detected in the community (Figure 2), resulting in a subversion of the dynamics of circulating lineages. At the end of 2022, BA.5 sub-lineages became the major circulating lineages in the local community. Interestingly, the overall case number did not significantly change. This might be attributed to the broad immunity driven by natural infection in the past few months. On the other hand, the relatively steady death rate could be explained by the weaker virulence of Omicron variants, which have been shown to cause less severe symptoms and be less fatal compared to other variants [13].

This study has its limitations. During the fifth wave, the mode of COVID-19 case detection changed. Many COVID-19 positive cases were confirmed by at-home rapid antigen tests (RAT) instead of being reported by public hospitals—the sample sources of this study. These patients were less likely to be encountered in the public healthcare system. Consequently, specimen collection from these patients was challenging, leading to a possible disparity between the figures obtained in the present study and the actual situation, as well as fluctuation in sample availability. To mitigate the sampling biases and the impact of sample quantity fluctuation, we ensured a minimum of 35 samples per month in the analysis. Second, the geographic regions of samples were not considered, while such information was critical in studying the spread pattern and dynamics of the viruses. Third, the genome of SARS-CoV-2 is prone to mutation, leading to inaccuracies of phylogenetic analysis. The effect of such nucleotide changes was minimized by several means, such as strict screening of high-quality sequences and the option --keeplength in MAFFT that helped to maintain the length of existing alignments by removal of insertions. The phylogenetic analysis result was also validated using other bioinformatic tools.

Our study suggested that the emergence of new lineage was inevitable unless the mandatory quarantine period was seven days or longer, as the highly infectious nature of Omicron variants made transmission difficult to contain. Additionally, such a rigorous quarantine policy resulted in a drastic decline in the number of passenger arrivals [14,15]. In the era of Omicron, the reduced virulence of the variants has provided an impetus for the resumption of the tourism industry and other economic activities. At the end of 2022, the government canceled all the social distancing measures and the restrictions and vaccination requirements for inbound travelers were lifted. However, molecular surveillance of COVID-19 remains essential to monitor the emergence of new recombinant variants that may bear significant genetic variations and to correlate their predominance in the community with the hospitalization rate and death rate to assess their virulence.

## 5. Conclusions

To conclude, this work characterized the dynamic changes in lineages in the local community in 2022. This study stands out from previous studies by utilizing thousands of in-house sequenced COVID-19 whole genomes, rather than solely relying on computational modeling or archived genomic data. By integrating genomic data with well-organized public health policy information, the study fosters valuable insights for future interdisciplinary collaborations in combating various contagious diseases. This combination allows for nearly real-time evaluation of the effectiveness of implemented public health policies, enhancing our ability to respond to epidemics.

## Figures and Tables

**Figure 1 viruses-16-00981-f001:**
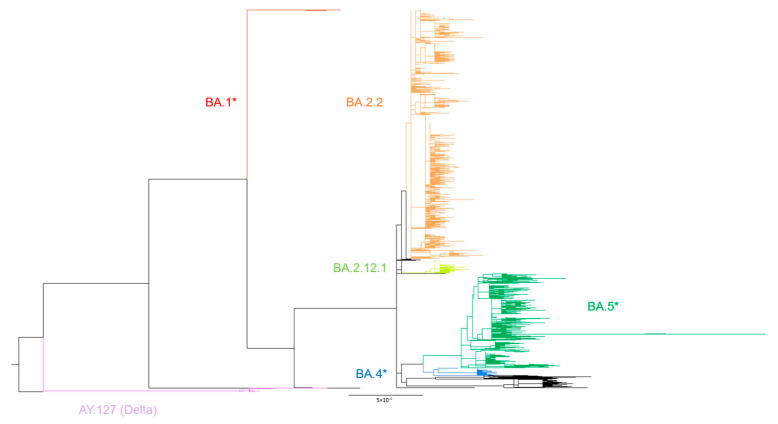
The phylogeny of the local cases was determined by IQ-TREE. Lineages were classified by pangolin v4.2. The scale bar indicates the genetic distance between sequences. AY.127-, BA.2.2- and BA.2.12.1-related cases are in lilac, orange and lime green, respectively. BA.1 sub-lineages (red), BA.5 sub-lineages (green) and BA.4 sub-lineages (blue) were grouped and represented by asterisk symbol respectively for clearer illustration.

**Figure 2 viruses-16-00981-f002:**
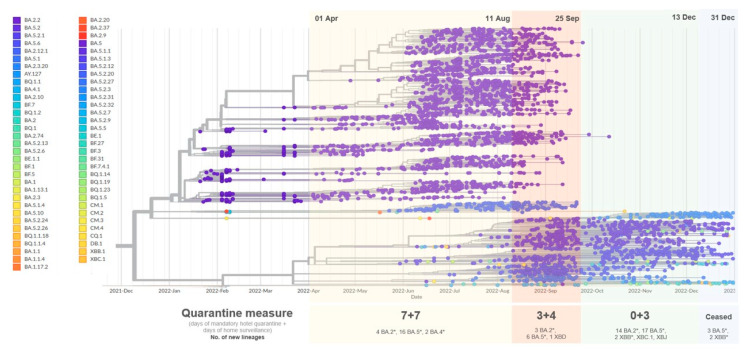
A time tree constructed using Nextstrain showing the dynamics of co-circulation of Omicron sub-variants in the community in 2022. Each strain was represented by a particular color. Wuhan-Hu-1 genome (NCBI accession number: MN908947.3) was used as the reference in the construction of this tree. Sub-lineages were represented by asterisk symbol for clearer presentation.

**Figure 3 viruses-16-00981-f003:**
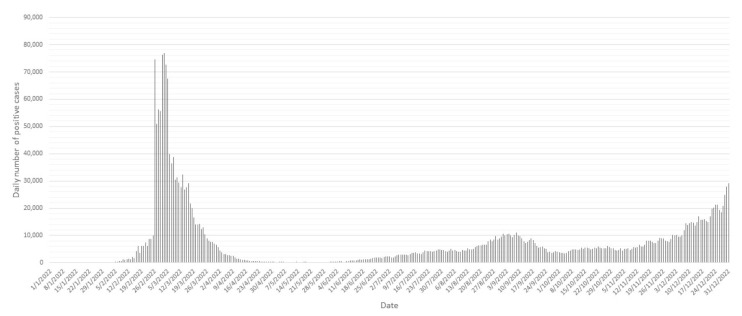
The total local infection cases during 2022. Data were adopted from DATA.GOV.HK.

**Figure 4 viruses-16-00981-f004:**
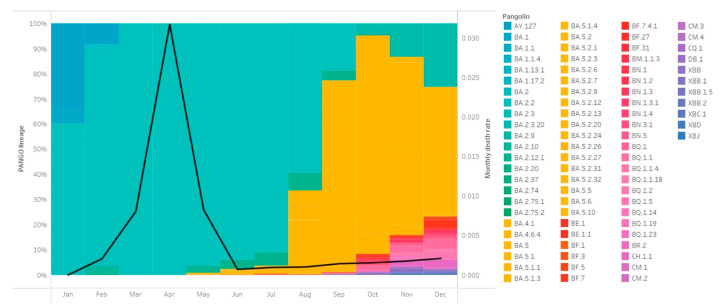
The proportion of circulating lineages and monthly death rate (line) in Hong Kong throughout 2022. The total case number and death case number were adopted from DATA.GOV.HK. Each color represents a particular lineage. The monthly death rate was calculated by dividing the monthly number of positive cases by the monthly number of death cases.

## Data Availability

The data that support the findings of this study are openly available in GISAID at https://gisaid.org/ (accessed on 28 April 2023). Accession numbers are listed in Appendix A.

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
