# Peer review of "Transmission Patterns of Co-Circulation of Omicron Sub-Lineages in Hong Kong SAR, China, a City with Rigorous Social Distancing Measures, in 2022"

_viruses, 2024, doi:10.3390/v16060981_

Round 1

Reviewer 1 Report

Comments and Suggestions for Authors

The conclusions of the present study are already well-known facts. What is the novelty of the current study? Please justify.

Reviewer 2 Report

Comments and Suggestions for Authors

Summary of the Work

The authors performed a statistical analysis to assess the correlation between changes in the number of circulating SARS-CoV-2 lineages and adjustments to quarantine arrangements. The local SARS-CoV-2 genomes were sequenced using the Oxford Nanopore GridION sequencer. The Oxford Nanopore GridION sequencer, MAFFT, and IQ-TREE have been used for sequencing the local SARS-CoV-2 genomes, to generate SARS-CoV-2 consensus genomes and to get the maximum likelihood phylogeny of these genomes, respectively.

The authors concluded that, since preventing the introduction and subsequent community spread of highly contagious Omicron variants is impossible, the best practice is to monitor the emergence of new recombinant SARS-CoV2 variants regularly.

General Comments

- In general, acronyms are not introduced in the abstract. In addition, all acronyms, even those well-known in the literature, must be specified when they are first mentioned in the manuscript. For instance, please specify MFFT, IQ-TREE, GrigION, etc.

(MAFFT = Multiple Alignment using Fast Fourier Transform;

IQ-TREE = Software for phylogenomic inference;

GridION = A benchtop real-time sequencing device for DNA and RNA sequencing, etc.).

- The limitations of the methods (software, etc.) used by the authors to investigate the sequenced 4,684 local SARS-CoV-2 genomes were not properly highlighted and discussed.

- The statistical analysis carried out by the authors was not clearly illustrated as the values of some key parameters were not reported.

- In my opinion, the conclusions and recommendations of this work (Section 5.) need to be rewritten. Summarized in this form, they appear quite obvious and therefore unattractive.

- The list of references is not exhaustive. It should also be completed taking into account the suggestions below.

The following suggestions are intended to help authors fill some gaps in the work.

Suggestions

1) MAFFT is a powerful tool for multiple sequence alignment. However, we know that it may encounter some limitations when used specifically for generating SARS-CoV-2 consensus genomes. We know that SARS-CoV-2 genomes can undergo genomic rearrangements such as recombination events, insertions, and deletions. MAFFT's alignment algorithms are primarily designed for aligning linear sequences and may not effectively handle complex genomic rearrangements, leading to inaccuracies in the consensus genome. For instance, SARS-CoV-2 genomes can vary in length due to insertions, deletions, and duplications. MAFFT's algorithms may face challenges in aligning sequences with substantial length variation, potentially resulting in errors or gaps in the consensus alignment. The authors are asked to discuss these issues.

2) SARS-CoV-2 sequencing data may contain ambiguous bases (e.g., Ns) representing uncertain nucleotides. We may object that MAFFT's handling of ambiguous bases in sequence alignment may affect the accuracy of the consensus genome, particularly if there are discrepancies or inconsistencies in the input data. The authors are asked to dispel this possible objection.

3) When dealing with SARS-CoV-2 genomes that exhibit variations in length due to insertions, deletions, and duplications, an alternative tool that may provide better handling of such complexities is MUSCLE (Multiple Sequence Comparison by Log-Expectation). MUSCLE is a reliable alternative to MAFFT for aligning SARS-CoV-2 genomes, especially when dealing with sequences that exhibit length variation due to insertions, deletions, and duplications. Its combination of accuracy, speed, and flexibility makes it well-suited for analyzing viral genomic data and generating high-quality alignments for downstream analysis.

3a) Have the authors found ARS-CoV-2 genomes that exhibit variations in length due to insertions, deletions, and duplications?

3b) In this situation, have the authors considered alternative methods to MAFFT, such as MUSCLE?

4) IQ-TREE is a powerful tool for phylogenetic analysis. However, this software has some limitations when used to analyze SARS-CoV-2 genomes. Indeed, IQ-TREE requires selecting an appropriate model of sequence evolution for phylogenetic analysis. While it provides various model selection criteria, choosing the best-fitting model for SARS-CoV-2 genomes can be challenging due to the complex evolutionary dynamics of the virus. The authors are asked to discuss this limitation by specifying the model selection criteria they have adopted to investigate the 4,684 local SARS-CoV-2 genomes sequenced in 2022 and the 83 SARS-CoV-2 lineages identified in 2022.

5) There is another important aspect that needs clarification. In general, the availability of SARS-CoV-2 genome sequences varies across geographic regions and periods.  This leads to potential sampling biases in the dataset. Biases in sampling can influence the inferred phylogenetic relationships and may affect the interpretation of evolutionary dynamics. Issues linked to the geographical regions (and time) have not been mentioned and discussed in this work. The authors are asked to fill this gap.

6) SARS-CoV-2 is known to undergo recombination, where genetic material is exchanged between different viral strains. Traditional phylogenetic methods may struggle to accurately capture recombination events, leading to potentially misleading tree topologies. To mitigate this limitation, it is customary to combine IQ-TREE with other bioinformatics tools such as tools like BWA, Bowtie2, HISAT2, and STAR or other popular phylogenetic analysis tools such as RAxML, PhyML, MrBayes, BEAST, and MEGA. Have the authors considered the possibility of integrating the IQ-TREE software with other bioinformatic tools to perform a more robust analysis of SARS-CoV-2 genomes?

7) Carrying out a correct and exhaustive statistical analysis is very important. The accuracy of phylogenetic inference depends largely on the quality of the input data. SARS-CoV-2 genomes may have sequencing errors, gaps, or other artifacts that can significantly affect the accuracy of phylogenetic analysis.

7a) According to the authors's findings, is the relationship between the variables monotonic?

7b) If yes, what are the values of the Pearson correlation coefficients?

7c) If not, what are the values of the Spearman rank correlation coefficient and Kendall's tau?

Conclusions

The work is interesting and current. However, it is vulnerable in several aspects. Authors are advised to take the above suggestions into account. It is also recommended to rewrite the conclusions (Section 5.). Written in this form, one could argue that we could have said the same things without having to resort to any model and the aid of any data analysis.

Comments on the Quality of English Language

English should be double-checked; several typos were found.

Round 2

Reviewer 1 Report

Comments and Suggestions for Authors

Authors responded to the reviewer's comments and updated the manuscript as well. 

Author Response

Thank you very much for taking the time to review this manuscript. 

Reviewer 2 Report

Comments and Suggestions for Authors

I carefully read the authors' replies to my comments and considered the suggestions. The authors have made a significant effort to clarify the points raised in my previous report. Section 5. ("Conclusions") has also been rewritten. In my opinion, this version of the manuscript may be published.

Author Response

(The authors gave the same response as above.)
